# Schottky-Diode Design for Future High-Speed Telecommunications

**DOI:** 10.3390/nano13091448

**Published:** 2023-04-24

**Authors:** Chi-Ho Wong, Leung-Yuk Frank Lam, Xijun Hu, Chi-Pong Tsui, Anatoly Fedorovich Zatsepin

**Affiliations:** 1Department of Industrial and Systems Engineering, The Hong Kong Polytechnic University, Hong Kong 999077, China; roych.wong@polyu.edu.hk; 2Research Institute for Advanced Manufacturing, The Hong Kong Polytechnic University, Hong Kong 999077, China; 3Department of Chemical and Biological Engineering, The Hong Kong University of Science and Technology, Hong Kong 999077, China; 4Institute of Physics and Technology, Ural Federal University, Yekaterinburg 620002, Russia

**Keywords:** dielectric properties, 2D materials, Schottky diode, energy harvesting system

## Abstract

The impact of 5G communication is expected to be widespread and transformative. It promises to provide faster mobile broadband speeds, lower latency, improved network reliability and capacity, and more efficient use of wireless technologies. The Schottky diode, a BN/GaN layered composite contacting bulk aluminum, is theoretically plausible to harvest wireless energy above X-band. According to our first principle calculation, the insertion of GaN layers dramatically influences the optical properties of the layered composite. The relative dielectric constant of BN/GaN layered composite as a function of layer-to-layer separation is investigated where the optimized dielectric constant is ~2.5. To push the dielectric constant approaching ~1 for high-speed telecommunication, we upgrade our BN-based Schottky diode via nanostructuring, and we find that the relative dielectric constant of BN monolayer (semiconductor side) can be minimized to ~1.5 only if it is deposited on an aluminum monolayer (metal side). It is rare to find a semiconductor with a dielectric constant close to 1, and our findings may push the cut-off frequency of the Al/BN-based rectenna to the high-band 5G network.

## 1. Introduction

5G communication promises to revolutionize wireless technology, enabling faster download and upload speeds, lower latency, more robust and reliable connections, and many more innovative remote applications [1], such as medical surgery and automobiles. With the potential for improved connectivity, 5G will enable more efficient, creative, and cost-effective solutions for businesses and consumers alike. It could also create opportunities for developing countries to gain similar access to high-speed internet as those in developed nations, providing a better quality of wireless communication. 5G wireless communication is based on harvesting wireless energy with the help of a local antenna array and a low-power automated transceiver [1]. The automated transceiver assigned the optimum frequency channels where the local antennas are connected to the LTE network. The frequency of the low-band 5G is close to 700 MHz, and the transmission of the mid-band 5G is about 2.5–3.7 GHz [2]. The high-band 5G frequencies of 25–40 GHz should provide the fastest internet speed of a few gigabits per second [3]. Although a frontier company such as Samsung has been investigating the feasibility of mobile communications at frequencies of 28 and 38 GHz [3], the current state-of-the-art Schottky diode harvests wireless energy up to 8 GHz–12 GHz (X band) only [4]. This X-band Schottky diode is made up of two-dimensional MoS_2_ sheets and bulk palladium^4^, where the relative dielectric constant of MoS_2_ sheets is above six [5]. Harvesting wireless energy beyond X-band requires the clearance of electric charges in the depletion region rapidly, where the cut-off frequency of the Schottky diode is inversely proportional to the dielectric constant [6]. A high dielectric constant increases the capacitance of Schottky diodes and eventually requires a longer transient time to clear the electric charges [7].

Designing a Schottky diode that is capable of harvesting wireless energy at the high-band 5G signals has proven to be challenging because of the issue of finding semiconductors that have a relative dielectric constant [7] close to 1. This is because of the problem of having high electric susceptibilities in semiconductors. For example, the dielectric constants of 3D silicon, 3D germanium, and 2D MoS_2_ sheets are 11, 16, and 6, respectively [7,8]. As the dielectric constant depends on size and pressure [9], we have decided to alter the nanostructure of a Schottky diode in order to create an ultra-low dielectric constant, which would better prepare for the 5G world. The dielectric constants of GaN and BN films are ~8 and ~3.5, respectively. We are going to study whether the heterostructure made up of these two materials will yield a much lower dielectric constant. In this paper, two Schottky diodes will be proposed: (BN-GaN/Al and BN monolayer/Al monolayer). We will apply the ab-initio simulations to design high-speed Schottky diodes with a key feature of ultralow dielectric constant.

## 2. Materials and Methods

The BN layers and GaN layers are stacked alternatingly, as shown in Figure 1. We apply geometric optimization under a boundary condition that the layer-to-layer distance remains at 0.5 nm. The geometric optimization is implemented under spin-restricted GGA-PBE, functional [10,11] under the CASTEP platform. The convergence tolerance is that the maximum displacement and the maximum force of atoms are 0.002 Å and 0.05 eV/Å, respectively. We calculate the band structure and the dielectric constant using the same DFT functional. The maximum SCF cycle is 100, and the SCF tolerance is 2 × 10^−6^ eV/atom. The internal of k-points is 1/Å. The ultrasoft pseudopotential is applied. The bulk aluminum is attached to BN/GaN layered composite side-by-side with specific crystallographic directions, i.e., the Al [100] and BN/GaN [1120] axes are parallel (abbreviated as Schottky diode A).

On the other hand, we examine the semiconducting properties of Schottky diode B, in which the boron nitride monolayer is deposited on top of an aluminum monolayer. After implementing geometric optimization in the presence of an aluminum monolayer under the same DFT method, we proceed to calculate the band gap and the dielectric constant (parallel to the surface) of the BN monolayer.

## 3. Results and Discussions

The band structure of the BN-GaN layered composite is plotted in Figure 2. Despite the band gap of the isolated BN sheet being as large as 6 eV, the direct band gap of the BN/GaN layered composite is reduced to 1.6 eV because of the proximity effect across the layers. The bigger gallium atoms raise the atomic density in each GaN layer so that a stronger internal pressure [12,13] is created in the composite. The insertion of GaN layers also strengthens electronic overlaps along the lateral direction [12,13]. The wavefunction of electrons overlapped more effectively in the BN/GaN layered composite, and hence a narrower band gap is observed. The theoretical band gaps simulated by the GGA functional like PBE or PW91 are usually underestimated [14,15,16,17]. However, there is no universal way to calibrate the band gap calculated by the ab-initio simulations unless the experimental band gap of the BN-GaN layered composite is known in combination with the parametric tests in the ab-initio simulations semi-empirically. Though our calculated band gap is slightly underestimated, the band gap of the BN/GaN layered composite still reaches 1.6 eV, in which the value of the band gap is suitable to be a part of the Schottky diode.

To identify if the BN/GaN layered composite belongs to a p-type or n-type semiconductor, we estimate the Fermi level (E_F_) relative to the conduction band (E_c_) and the valence band (E_v_), respectively. Figure 2 shows that the Fermi level of the BN/GaN layered composite is located closer to the conduction band, and we conclude that the BN-GaN layered composite is an n-type semiconductor, where the energy levels before the bulk Al and BN-GaN join together, drawn in Figure 3. The Fermi level is located nearer to the conduction band because the bigger gallium atom decreases the positive charge density of the composite [18].

Figure 3 displays the simplified energy diagrams of Schottky diode A. The work function of aluminum is 4.1 eV, and the electron affinity of the BN/GaN layered composite is 3 eV. The Schottky barrier refers to the offset between the work function of the metal and the electron affinity of the semiconductor [7,18]. The Schottky barrier of most silicon-based diodes is around 0.7 eV [19]. By aligning the energy states relative to the vacuum level, the Schottky barrier of Schottky diode A is 1.1 eV. In other words, the driven voltage of Schottky diode A for forward-bias operation is comparable to the modern silicon-based diodes [7].

The dielectric function describes the response of the material subjected to a time-dependent electromagnetic field [18]. The excitations induced by the interaction of incident electrons with materials refer to the real and imaginary parts of the dielectric function [20]. We show the relative dielectric constant of the BN/GaN layered composite in Table 1. The relative dielectric constant of the BN/GaN layered composite is as low as 2.5. In comparison with the modern X-band rectenna consisting of two-dimensional MoS_2_ sheets and palladium [4], the dielectric constant of BN/GaN layered composite is at least 50% lower than the dielectric constant of two-dimensional MoS_2_ sheets [5]. With the use of the BN/GaN layered composite as a part of the Schottky diode, it is possible to push the cut-off frequency far beyond X-band because the cut-off frequency is inversely proportional to the capacitance across the depletion region. The non-regular atomic layers across the metal-semiconductor interface make the theoretical prediction of junction resistance inaccurate. However, the electrical resistance of aluminum is ~4 times smaller than that of palladium, and hence a low junction resistance is likely expected. The imaginary parts of the dielectric constants of Schottky diodes A and B are close to zero, and therefore their phase lags are likely negligible [20].

The dielectric constant is directly proportional to the permittivity of the medium [18]. If the layer-to-layer distance of the BN/GaN layered composite increases to 0.8 nm, the relative dielectric constant can drop to 2.6, but the band gap rises to 2.5 eV, as shown in Figure 4. Materials with low dielectric constants generally have weaker intermolecular interactions, meaning their molecules are less likely to hold onto, and may facilitate the flow of, the electric charge. Hence, weakening the layer-to-layer coupling decreases the effectiveness of electric polarization in the medium [20] and presumably drops the permittivity of the substance. While softening the layer-to-layer coupling may yield a low dielectric constant, we proceed to study the dielectric properties of the isolated BN monolayer. We observe that the relative dielectric constant of the isolated BN monolayer can drop even further to 2.4.

The BN layer is used as a building block because the dielectric constant of pristine hexagonal BN film is already small (~3.5). The relationship between the dielectric constant $\varepsilon_{r}$ and imposed E-field^6^ is $\varepsilon_{r}=1+\frac{2\pi B\gamma^{1/2}}{\xi^{2}}\times f(Aiya)$, where B is related to the optical absorption in the absence of static field. $\gamma \propto \frac{1}{F_{field}}$, where $F_{field}$ is the force under an imposed E-field [6]. f(Airy) is the integral related to Airy functions [6]. To impose an E-field ($\gamma \propto \frac{1}{F_{field}}$) can drop the dielectric constant of semiconductors [6], we need to insert another thin-film semiconducting material with the same hexagonal structure to create an internal E-field perpendicular to the plane, where the GaN layer is preferable. By introducing a dissimilar size of cation (Ga and B) between the adjacent layers, the dissimilar layers contribute an internal E-field perpendicular to the plane. Hence, the closely packed BN and GaN layered composite yields a lower dielectric constant than either BN or GaN layers where the positive charge density of B ion is higher than that of Ga ion in the layers, and meanwhile, the B ion induces negative charges on the nearest GaN layers under an induced internal E-field [6]. The electric field usually drops the dielectric constant by reducing the number of effective charge carriers in the material. This happens because the polar molecules in the material are oriented and shifted in response to the external E field. This impedes materials from concentrating the electrostatic flux. As a result, the induced negative charges on GaN layers owing to the induced E-field reduce the dielectric constant [6] in the entire composite. The band gap is increased from 1.6 eV to 2.5 eV because the wider layer-to-layer separation reduces the internal pressure of the composite along the lateral plane [12,18].

Motivated by the benefits of the internal E-field between the layers [6], we decided to examine if the application of an internal E-field between an aluminum monolayer and a BN monolayer can drop the dielectric constant of the BN monolayer below 2.0. After the BN monolayer is relaxed in the presence of an aluminum monolayer, the B-N-B bond angle changes from 120 degrees to 116.9 degrees, and the N-B bond length changes from 1.45 Å to 1.48 Å. The Al ions in the aluminum monolayer create a high positive charge density and induce negative charges on the BN monolayer, which sets up an internal E-field. The induced E-field drops the dielectric constant^6^ of the BN monolayer below 2.0 successfully. As listed in Table 1, the boron nitride monolayer in Schottky diode B shows the relative dielectric constant of 1.5 only, and the energy levels of the components are listed in Figure 5. In comparison with bulk aluminum, the work function of the aluminum monolayer is only 2.5 eV because the size-dependent work function is always expected [21]. We draw a parallel between the function parameters of Schottky diode B and the MoS_2_-based Schottky diode in Figure 6. Although the energy barrier of Schottky diode B is increased to 1.41 V, its ultralow dielectric constant is still able to harvest wireless energy beyond X-band communication if the driven voltage is above the forward-bias voltage [7]. The dielectric constant of the BN monolayer in Schottky diode B is ~4 times smaller than the dielectric constant of two-dimensional MoS_2_ sheets in the modern X-band Schottky diode [4]. As the cut-off frequency of the Schottky diode is inversely proportional to the capacitance across the depletion region [7], the ultralow dielectric constant of Schottky diode B may be able to boost the cut-off-frequency towards the high-band 5G signals. Moreover, the wide band gap in the Schottky diode B is expected to operate at higher temperatures than the MoS_2_-based Schottky diodes because a wide band gap in a semiconductor allows for greater resistance to thermal breakdown, as the material is able to withstand higher temperatures without carrier excitation. This increases the maximum operating temperature of the Schottky diode, allowing it to operate more efficiently at higher temperatures. Concerning electric conductivity, aluminum is more metallic than palladium, which allows a low ohmic resistance value (minimizes power dissipation) on the metallic side.

## 4. Conclusions

Our research shows the potential for utilizing a BN/GaN layered composite and aluminum to build Schottky diodes for creating high-frequency wireless energy-harvesting devices. By tuning the induced electric field of the interface, we identify that the dielectric constant of the boron nitride monolayer on top of the aluminum monolayer is as low as 1.5, suggesting its suitability for harvesting high-band 5G signals. The implication of this finding is that the proposed heterostructures are promising to construct devices capable of harvesting higher frequency signals, such as those produced by 5G networks. Such a device would enable the capture and conversion of these high-frequency signals into energy sources, creating an efficient form of energy transfer and high-speed telecommunications.

## Figures and Tables

**Figure 1 nanomaterials-13-01448-f001:**
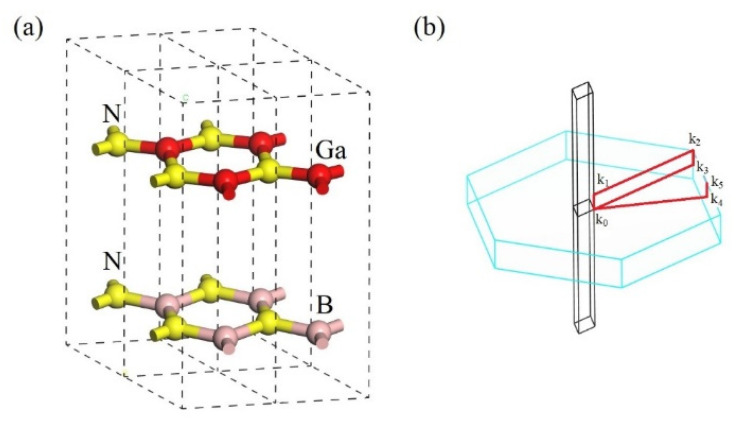
(**a**) The repeated unit of BN/GaN layered composite. (**b**) The real (black) and reciprocal (light blue) spaces. The red lines label the direction of k space.

**Figure 2 nanomaterials-13-01448-f002:**
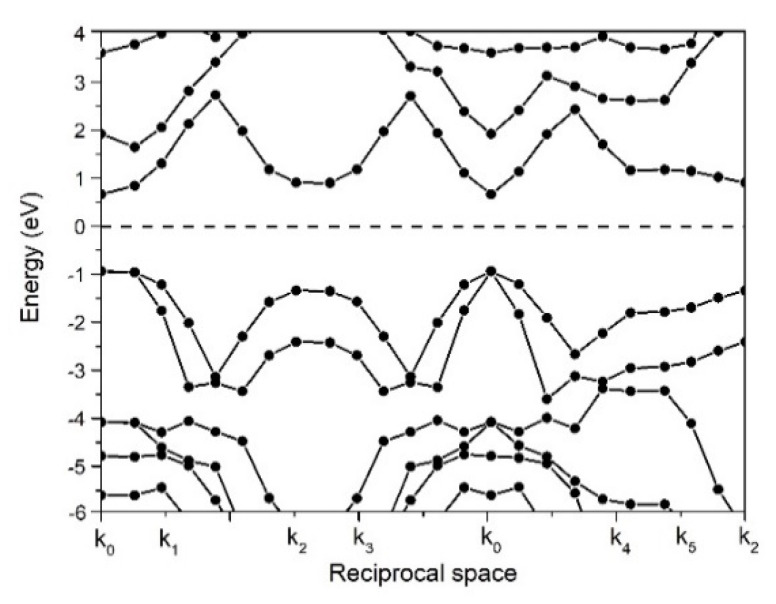
The band structure of BN/GaN layered composite. The layer-to-layer distance is 0.5 nm. The Fermi level is shifted to 0 eV for convenience.

**Figure 3 nanomaterials-13-01448-f003:**
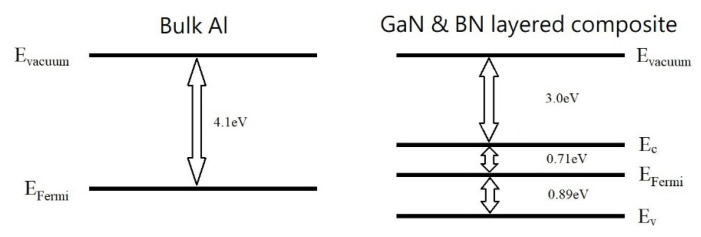
The energy levels of the bulk Al and the BN-GaN layered composite before they join together to form the Schottky diode A. The work function is the energy difference between the vacuum level E_vaccum_ and Fermi level E_Fermi_, while the band gap is the energy difference between the conduction E_c_ and valence bands E_v_. The electron affinity is the energy difference between the vacuum level E_vaccum_ and conduction band E_c_. Subtracting the work function from the electron affinity yields the Schottky barrier.

**Figure 4 nanomaterials-13-01448-f004:**
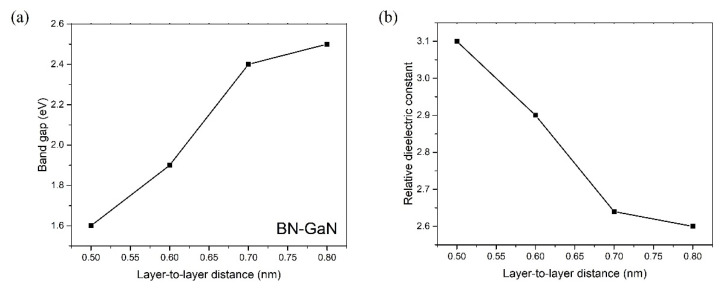
(**a**) The band gap and (**b**) the dielectric constant of the BN-GaN composite depend on the layer-to-layer separation d between BN and GaN layers.

**Figure 5 nanomaterials-13-01448-f005:**
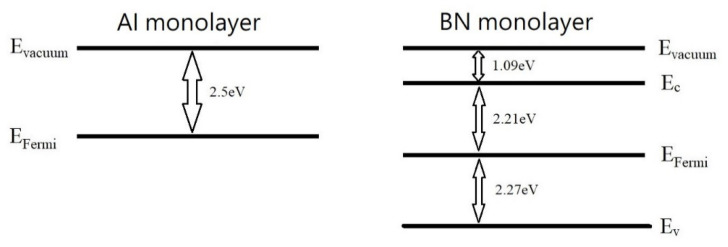
The energy levels of the Al monolayer and BN monolayer before they join together to form Schottky diode B. **Left panel**: The work function is defined by E_vaccum_—E_Fermi_; **Right panel**: The band gap is defined by E_c_—E_v_ and the electron affinity is defined by E_vaccum—_E_c_; The work function subtracted by the electron affinity is called the Schottky barrier.

**Figure 6 nanomaterials-13-01448-f006:**
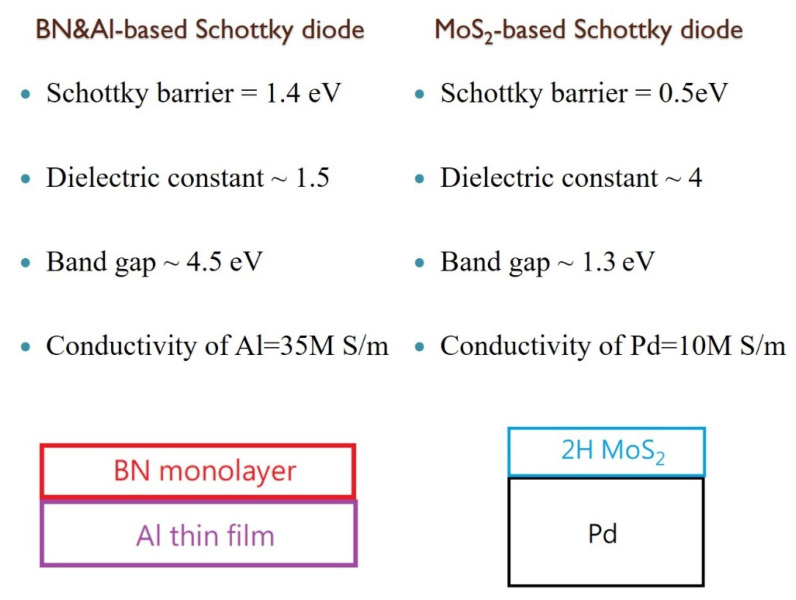
Comparison between Schottky diode B and the MoS_2_-based Schottky diode [4]. The electric conductivity refers to 293 K.

**Table 1 nanomaterials-13-01448-t001:** The dielectric constants of Schottky diodes A and B.

	Material (s)	Dielectric Constant (s)
Schottky diode A	Bulk Al and BN/GaN layered composite	2.5–3.1 (d: 0.5 nm–0.8 nm)
Schottky diode B	Al monolayer and BN monolayer	1.5

## Data Availability

Data are sharable under a reasonable request.

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
