# Peer review of "Schottky-Diode Design for Future High-Speed Telecommunications"

_nanomaterials, 2023, doi:10.3390/nano13091448_

Round 1

Reviewer 1 Report

I have completed my review of the manuscript, titled "nanomaterials-2345546", and I would like to offer some comments to help improve the quality of the manuscript. In order to enhance the overall quality of the manuscript, please address the following points in the revised version:

  1. In the abstract section, it is important to clearly indicate the novelty of your work to capture the reader's attention and highlight the significance of your research.

  2. Ensure that there is the consistent spacing between numerical values and units throughout the manuscript. This will ensure that the text is presented in a clear and concise manner.

  3. In the introduction section, it is essential to provide a more detailed explanation of the motivation behind your work in order to help readers better understand the significance and potential impact of your research.

  4. The legends for your figures, particularly figures 3 and 5, should be expanded to provide a more comprehensive understanding of the data being presented.

  5. Finally, the conclusion should be rewritten to effectively summarize the key findings of the manuscript and provide a clear and concise overview of the significance of your research.

I hope that these comments will be helpful in improving the quality of manuscript.

Moderate editing of English language required.

Author Response

I have completed my review of the manuscript, titled "nanomaterials-2345546", and I would like to offer some comments to help improve the quality of the manuscript. In order to enhance the overall quality of the manuscript, please address the following points in the revised version:

  • In the abstract section, it is important to clearly indicate the novelty of your work to capture the reader's attention and highlight the significance of your research.

Answer: Thank you for your comment. In the revised version, I have updated the abstract by adding “The impact of 5G communication is expected to be widespread and transformative. It promises to provide faster mobile broadband speeds, lower latency, improved network reliability and capacity, and more efficient use of wireless technologies.”

  • Ensure that there is the consistent spacing between numerical values and units throughout the manuscript. This will ensure that the text is presented in a clear and concise manner.

Answer: Thank you for your comment. I have added the spacing in the revised version.

  • In the introduction section, it is essential to provide a more detailed explanation of the motivation behind your work in order to help readers better understand the significance and potential impact of your research.

Answer: Thank you for the suggestions. In the revised paper, we have added “5G communication promises to revolutionize wireless technology, enabling faster download and upload speeds, lower latency, more robust and reliable connections, and many more innovative remote applications1 such as medical surgery and automobiles. With the potential for improved connectivity, 5G will enable more efficient, creative, and cost-effective solutions for businesses and consumers alike. It could also create opportunities for developing countries to gain access to the high-speed internet as those in developed nations, providing a better quality of wireless communication.” to highlight the significance of 5G communication”.

  • The legends for your figures, particularly figures 3 and 5, should be expanded to provide a more comprehensive understanding of the data being presented.

Thank you for your comment. We have expanded the font size of the legend in Figure 3 and 5. We have also updated the figure captions in Figure 3 and 5. 

  • Finally, the conclusion should be rewritten to effectively summarize the key findings of the manuscript and provide a clear and concise overview of the significance of your research.

Thank you for your comment. We have edited the conclusion in the revised version.

I hope that these comments will be helpful in improving the quality of manuscript.

  1. Moderate editing of English language required.

Answer: Thank you for your comment. We have edited it in the revised version

Reviewer 2 Report

Nanostructuring and heterogenization of thin films are key routes to obtain their desired optical properties by controlling their dielectric features. The authors here upgrade the BN-based Schottky diodes via nanostructuring and the formation of BN/GaN semiconductor hetero-interface on the metallic Al monolayer. The result suggests that the final dielectric constant can be ~1.5, near to 1.0, induced by the interface E field of the formed Schottky-diode  under the external E field , which is promising for high-band 5G communication. This result is good and the article is written well, which is deserved for our journal. There are some suggestions for authors to consider to improve this article before acceptance.

1.      Please give the detailed structure (Figure or scheme) and function parameters(Table) description of current MoS2-sheet based Schottky-diodes and analyze their features and issues. Better to compare their results with the current ones in the results and discussion section, including data in Table 1.

2.      Please give the detailed calculation procedure using DFT as support information.

3.      Please give the key reason to select the BN/GaN semiconductor as constructing materials, and discuss the fundamental mechanism to reduce the dielectric constant for the enhanced performance (better to show the equations) and the contribution of each layer.

4.      All the figures should use the high-resolution and the original magnified ones for easy read.

5.      These materials are very easy to be fabricated. All fabrication procedures are matured. It is better to do the experiment to confirm their calculation.

Author Response

Nanostructuring and heterogenization of thin films are key routes to obtain their desired optical properties by controlling their dielectric features. The authors here upgrade the BN-based Schottky diodes via nanostructuring and the formation of BN/GaN semiconductor hetero-interface on the metallic Al monolayer. The result suggests that the final dielectric constant can be ~1.5, near to 1.0, induced by the interface E field of the formed Schottky-diode under the external E field , which is promising for high-band 5G communication. This result is good and the article is written well, which is deserved for our journal. There are some suggestions for authors to consider to improve this article before acceptance.

  1. Please give the detailed structure (Figure or scheme) and function parameters(Table) description of current MoS2-sheet based Schottky-diodes and analyze their features and issues. Better to compare their results with the current ones in the results and discussion section, including data in Table 1.

Thank you for your comments. In the revised version, we have added Figure 6 to compare the Schottky diodes. We have also analyzed their features in Table 1 and Figure 6. In the Results and Discussion section, we have enriched the section by discussing the thermal issues and power dissipation…etc on page 6 in the revised manuscript.  

  1. Please give the detailed calculation procedure using DFT as support information.

Thank you for your comment. We have added the computational details in the revised version (page 2)

  1. Please give the key reason to select the BN/GaN semiconductor as constructing materials, and discuss the fundamental mechanism to reduce the dielectric constant for the enhanced performance (better to show the equations) and the contribution of each layer.

Thank you for your comment. We have added the key reason on page 4-5 in the revised version (The BN layer is used as a building block because the dielectric constant of pristine hexagonal BN film is already small (~3.5). While E-field can drop the dielectric constant of semiconductors, we need to insert another thin-film semiconducting material with the same hexagonal structure to create an internal E-field perpendicular to the plane, where the GaN layer is preferable. By introducing a dissimilar size of cation (Ga & B) between the adjacent layers, the dissimilar layers contribute an internal E-field perpendicular to the plane.) We have also added the equations on page 4 in the revised paper.

  1. All the figures should use the high-resolution and the original magnified ones for easy read.

Thank you for your suggestions. We have updated the resolution of the figures in the revised version. We have also uploaded the original JPEG files to the editors.

  1. These materials are very easy to be fabricated. All fabrication procedures are matured. It is better to do the experiment to confirm their calculation.

Thank you for your comment. I work in my department as a theorist and the nature of this project is theoretical investigation. However, I will apply for grants and contact some experimental partners to verify the simulation results. Hence, if possible, please allow us to do the experimental task as future work.

Reviewer 3 Report

This paper reports the design of a Schottky diode for 5G wireless communication, based on first principles, to verify that the insertion of gallium nitride layer greatly affects the optical properties of the layered structure and greatly reduces its dielectric constant. The work sounds very interesting and meaningful, and the analysis is quite clear, but there are still some concerns about this article, so the paper needs to be minor revised and the following points need to be addressed

1In Computational Details, The author should provide more calculation details to enhance the persuasiveness of the article.

2. The format of the article should be modified as required.

The author should polish the language to improve the overall quality of the article.

Author Response

This paper reports the design of a Schottky diode for 5G wireless communication, based on first principles, to verify that the insertion of gallium nitride layer greatly affects the optical properties of the layered structure and greatly reduces its dielectric constant. The work sounds very interesting and meaningful, and the analysis is quite clear, but there are still some concerns about this article, so the paper needs to be minor revised and the following points need to be addressed:

1、In Computational Details, The author should provide more calculation details to enhance the persuasiveness of the article.

Thank you for your suggestions. We have added the computational details on page 2 in the revised manuscript.

  1. The format of the article should be modified as required.

Thank you for your comment. We have modified the acknowledgment, the format of author contributions, conflict of interest…etc in the revised manuscript. We also removed the numbers before the introduction, materials and methods, results and discussion, and conclusion.  

  1. Comments on the Quality of English Language

The author should polish the language to improve the overall quality of the article.

Thank you for your comment. We have edited it in the revised paper

Round 2

Reviewer 3 Report

This paper reports the design of a Schottky diode for 5G wireless communication, based on first principles, to verify that the insertion of gallium nitride layer greatly affects the optical properties of the layered structure and greatly reduces its dielectric constant. The work sounds very interesting and meaningful, and the analysis is quite clear, but there are still some concerns about this article, so the following points need to be addressed

1In Computational Details, The author should provide more calculation details to enhance the persuasiveness of the article.

2. The format of the article should be modified as required.

The author should polish the language to improve the overall quality of the article.